# Electrically Conductive Adhesive Based on Thermoplastic Hot Melt Copolyamide and Multi-Walled Carbon Nanotubes

**DOI:** 10.3390/polym14204371

**Published:** 2022-10-17

**Authors:** Paulina Latko-Durałek, Michał Misiak, Anna Boczkowska

**Affiliations:** 1Faculty of Materials Science and Engineering, Warsaw University of Technology, Wołoska 141 Street, 02-507 Warsaw, Poland; 2Technology Partners Foundation, Pawińskiego 5A Street, 02-106 Warsaw, Poland

**Keywords:** hot melt adhesive, carbon nanotubes, adhesion, rheology, microstructure

## Abstract

For the bonding of the lightweight composite parts, it is desired to apply electrically conductive adhesive to maintain the ability to shield electromagnetic interference. Among various solvent-based adhesives, there is a new group of thermoplastic hot melt adhesives that are easy to use, solidify quickly, and are environment-friendly. To make them electrically conductive, a copolyamide-based hot melt adhesive was mixed with 5 and 10 wt% of carbon nanotubes using a melt-blending process. Well-dispersed nanotubes, observed by a high-resolution scanning microscope, led to the formation of a percolated network at both concentrations. It resulted in the electrical conductivity of 3.38 S/m achieved for 10 wt% with a bonding strength of 4.8 MPa examined by a lap shear test. Compared to neat copolyamide, Young’s modulus increased up to 0.6 GPa and tensile strength up to 30.4 MPa. The carbon nanotubes improved the thermal stability of 20 °C and shifted the glass transition of 10 °C to a higher value. The very low viscosity of the neat adhesive increased about 5–6 orders of magnitude at both concentrations, even at elevated temperatures. With a simultaneous growth in storage and loss modulus this indicates the strong interactions between polymer and carbon nanotubes.

## 1. Introduction

Electrically conductive adhesives (EACs) are a group of materials developed to be used as adhesives in electronics or the aviation and automotive industry as a bonding medium of thermoplastic or thermosetting matrix composites. In the first application, the ECAs can replace the traditional Pb-Sn solder used to assemble the components of printed circuit boards. The high interest in ECAs in electronic packaging is due to easier processing and higher resolution printing, lower processing temperature, and environmental friendliness compared to Pb-Sn soldiers [1]. In the second application area, ECAs can improve electromagnetic shielding properties or lightning strike protection by forming the interlayer between the joining lightweight composites used in automotive or aircraft sectors. Here, the ECAs are a promising candidate for eliminating commonly used rivets resulting in lower stress concentration and weight of the final parts [2].

The main components of ECAs are polymer and electrically conductive fillers. The polymer matrix consists of both thermoplastics like polyimide and, mainly, thermosets such as silicone, epoxy, or acrylate resins, and they are responsible for the mechanical properties of the adhesive layer [3]. Because polymers (excluding conductive polymers) are insulators, they need to be modified with electrically conductive fillers, which provide a sufficient level of conductivity and do not decrease the overall mechanical performance. Of the broad range of conductive fillers, micro silver is most commonly used in the form of powder, flakes, spheres, nanowires, or dendrites [4]. However, to achieve a sufficient level of conductivity, it is necessary to add a high amount of silver–between 25 wt% and even 80 wt%. Such an amount causes difficulties during processing, high cost, and impairs the adhesive layer’s mechanical properties. Moreover, the electrical conductivity diminishes as an effect of corrosion and oxidation of the metallic filler and localization of the charge carriers [5,6]. To overcome these weaknesses, the new approach focuses on applying carbon-based fillers or nanofillers. The best-known examples include carbon nanotubes (CNTs), graphene, carbon black, and graphite. Due to their high surface area, they form a conductive network in the polymer at lower concentrations than metal fillers [7]. Moreover, they are light and not corroding, and their price is decreasing yearly owing to increased production. However, the critical challenge for the ECAs containing carbonaceous fillers is to achieve the percolated network that allows transferring electrons through the contact points between conductive fillers or by the tunneling effect. Carbon-based nanofillers form agglomerates, which must be destroyed during processing to achieve homogenous dispersion and distribution in the polymer matrix. Ideally, they will form the percolated conductive network at a low concentration [8]. However, it should be noted that the main parameter affecting the overall conductivity of the ECAs is the filler-filler contact resistance, which disturbs the free flow of the electrons. Therefore, agglomerates or microparticles with fewer contact points can positively affect the ECAs’ conductivity [1,9]. Hence, researchers analyze the mutual impact of both types of fillers–metallic and carbonaceous because the built network results in more effective conductance. For example, Luo et al. [10] proved that adding 4.5 wt% CNTs together with 50 wt% of silver flakes improved the electrical conductivity by about 255% compared to the sample without CNTs, and the values reached were 2.56 × 10^5^ S/m. The other tested possibility is to modify the surface of CNTs with silver to obtain functionalized CNTs and then add them to the polymer. However, such a solution resulted in a relatively low electrical conductivity of 2.83 × 10^−6^ S/m [11]. Instead of using conductive fillers, the ECAs can also be produced by modifying the polymer matrix with intrinsically conductive polymers such as polyaniline or polypyrrole [12].

Applying ECAs in electronics requires high electrical conductivity and good mechanical performance, high thermal conductivity, long curing time, resistance to low and high temperatures, and humidity [13]. For the other application of ECAs, such as adhesive bonding of the lightweight composite structures, the essential property is electrical conductivity needed for electromagnetic interference (EMI) shielding (0.1–10 S/m) or lightning strike protection (min. 10 S/m). The other mentioned requirements are high strength, the ability to withstand a wide temperature range, and a low coefficient of thermal expansion [14]. Furthermore, such ECAs should prevent delamination caused by insufficient adhesion between the layers and be easy to apply. Among various polymers used as a matrix in ECAs, the ones that have achieved the highest popularity are solvent-based epoxy, silicones, or acrylates. However, in recent years, solvent-free and thermoplastic hot melt adhesives have attracted great attraction. They are complex materials consisting of: (i) polymer (polyamide, polyurethane, polyolefins, ethylene copolymers) responsible for strength and hot tack; (ii) resin that improves contact with the substrate; (iii) tackifier that adjusts the glass transition and dilutes polymer; (iv) wax that increases setting speed [15]. Hot melts are available in the form of pellets, powder, sticks, and bars, which are solid at room temperature. When the temperature rises, they become liquid and soft, but they solidify during cooling. Depending on the formulation, hot melt adhesives offer a broad range of properties, including adhesion strength, working temperatures, and viscosity level [16]. Hot melts are used primarily where the process speed matters, such as non-wovens in sanitary products, construction, packaging, bottle labeling, bookbinding, or temporary attachments [17]. Hot melt adhesives have also found a place in the composites industry to improve the bonding strength of hybrid parts (different types of glass/carbon composites) [18]; improve the interlaminar toughness in lightweight composites [19]; increase the adhesion between the nanofibrous mat and its supporting woven polyester fabric [20] or as the conductive adhesive to provide an electrically conductive layer between bonding parts. So far, there are only a few publications describing ECAs based on hot melt adhesives, such as polyurethane [21], polyolefines [22], and our previous work on polyamide-based hot melts [23]. Compared to epoxy adhesives, the main advantages of the hot melts in adhesive bonding are lack of solvent, no requirement of special surface treatment; shorter curing time; lack of weight increase compared to rivets, and recyclability [18].

This study aimed to analyze the effect of the high loading of multi-walled carbon nanotubes (MWCNTs) on the properties of commercial non-conductive hot melt polyamide. For this research, a hot melt with very low viscosity was selected to examine how much the electrical conductivity will be improved, but also considering the rheological, thermal, and mechanical properties. Based on the achieved conductivity and the strength of the adhesive bonds, they are a promising candidate to be applied as ECAs for the adhesive bonding that requires EMI shielding.

## 2. Materials and Methods

### 2.1. Materials

As the hot melt adhesive, the copolyamide (coPA) hot melt in the form of granules with the trade name Vestamelt^®^722 was supplied by Evonik (Essen, Germany). According to the datasheet, the melting temperature is 107 °C and melt flow index = 310 g/10 min (2.16 kg/160 °C) classified that polymer as being low viscous. MWCNTs with the trade name NC7000 from Nanocyl (Sambreville, Belgium) synthesized by catalytic carbon deposition from the gas phase were used as the electrically conductive filler. The average diameter of that type of MWCNTs is 9.5 nm, length 1.5 µm, and purity > 95%.

### 2.2. Composites Fabrication

The selected coPA was mixed with 5% and 10% by weight (wt%) of MWCNTs using an industrial twin-screw extruder by Nanocyl. The pellets of neat coPA and both masterbatches were dried before each process in a vacuum oven at 60 °C for a minimum of 6 h. The specimens for the rheological and mechanical tests were prepared directly from the pellets using the laboratory injection moulding machine HAAKE Mini Jet Pro Piston Injection Molding System (ThermoScientific, Karlsruhe, Germany). The parameters of the injection moulding process are presented in Table 1.

### 2.3. Characterization Techniques

Firstly, the dispersion and distribution of MWCNTs in the masterbatches were examined using a Polarized Light Microscope (Bipolar-PL, PZO, Warsaw, Poland). Slides with a thickness of 2–3 μm were cut directly from the masterbatch pellets using an ultramicrotome (EM UC6, Leica, Vienna, Austria). From the obtained images (7 for each material) the number of MWCNT agglomerates was counted using image software (ImageJ). The percentage ratio AA was calculated by dividing the area of all agglomerates by the total area, excluding those agglomerates with a diameter lower than 1 µm. The second method applied to analyze the dispersion of MWCNTs was a Transmission Scanning Electron Microscope (HR STEM S5500, Hitachi, Tokyo, Japan) that allows the observations on a nanometer scale. For the analysis, the slides with a thickness of 80–90 nm were cut with diamond knives at −100 °C by an ultramicrotome (EM UC6, Leica, Vienna, Austria). The observations were performed at 30 kV.

An ARES rheometer (model 4400-0107, TA Instruments, New Castle, DE, USA) was used to analyze the viscoelastic properties of the hot melt adhesives by oscillatory test in the plate-to-plate mode. Firstly, a dynamic strain sweep test as a function of the variable strain γ (0.07–100%) at a constant frequency of 1 Hz was performed to determine the strain within the elastic range. Then a variable frequency test was performed with the determined deformation, in this case, 0.1% and at three temperatures of 180 °C, 200 °C and 220 °C. The specimens for the rheological analysis were produced by injection moulding in the form of rounds 25 mm in diameter and with a thickness of 1 mm.

The thermal stability of the materials was determined by thermogravimetric analysis (TGA). The examination was conducted on a TGA Q500 instrument (TA Instruments, New Castle, DE, USA). For this, samples of 10 ± 0.5 mg were prepared, then transferred to tared platinum pans and heated from 20 °C to 900 °C at a heating rate of 10 °C/min. Two flow rates of nitrogen of 10 mL/min in the chamber and 90 mL/min in the oven were used during the analysis. Thermal stability was determined by degradation temperatures of 5% (T_5%_) and 10% (T_10%_) weight loss, as well as by the maximum degradation peak (T_d_).

The thermal properties of all materials were examined by Differential Scanning Calorimetry (DSC) using the Q1000 Differential Scanning Calorimeter (TA Instruments, New Castle, DE, USA). The 6.5 ± 0.5 mg samples were placed in an aluminum hermetic crucible and analyzed under the heat-cool-heat program from −80 °C to 220 °C with a heating/cooling rate of 10 °C/min under a nitrogen atmosphere. The curves obtained from the test were analyzed using TA Universal Analysis 2000 software version 4.5A. The glass transition temperature (T_g_), melting temperature (T_m_), and enthalpy of melting (ΔH_m_) were determined from the heating curves, while the crystallization temperature (T_c_) and enthalpy of crystallization (ΔH_c_) were taken from the cooling curves. Due to the lack of data about the enthalpy of melting of 100% crystalline coPA, the degree of crystallinity was not calculated.

The electrical volume conductivity of unfilled and filled coPA was measured by the Keithley 6221/2182A device equipped with copper electrodes. Test samples with dimensions of 10 cm × 10 cm were prepared by pressing the pellets on a hydraulic press (Hydraulische Werkstattpresse WPP 50 E, Unicraft, Hallstadt, Deutschland) at the temperature of 115 °C and pressure of 30 MPa. Afterwards, small specimens of around 1 cm × 1 cm were cut from different sections of the bigger sample, to determine the homogeneity of the electrical conductivity. To maintain good contact between electrodes and the measured sample silver paste was applied.

Mechanical properties of the studied materials were analyzed by uniaxial tensile tests according to PN EN ISO527 using an MTS QTest 10 (MTS Systems, Eden Prairie, MN, USA) testing machine. Five test specimens represented each material. Tensile experiments were performed at a constant crosshead speed of 10 mm/min using an extensometer with a gauge length of 50 mm for strain measurements. From the stress-strain curves, the tensile modulus of elasticity, ultimate tensile strength (F_tu_) and elongation at break (ε_b_) were determined. Tensile tests were carried out on small dog bone specimens.

The hardness was determined based on the Shore method according to PN-EN ISO 868:2005 standard. The hardness test was carried out using the WHS-180 hardness tester by Wilson-Wolpert (ATM Qness, Mammelzen, Germany). All measurements were made using the Shore D scale for testing hard plastics. For each material, 25 measurements were made. The final result for coPA and its masterbatches containing MWCNT was obtained by calculating the arithmetic mean of the values. A load of 50 N was applied during the test.

The wettability of the surfaces of coPA and its masterbatches was tested by measuring the contact angle using an OCA15 (DataPhysics Instruments, Filderstadt, Germany) goniometer equipped with OCA software. The contact angle hysteresis was determined by calculating the difference between the advancing and receding contact angles. The test was performed on the round specimens from the injection moulding using a 5 μL droplet. The contact angle results are the averaged values of five different measuring points on each surface.

The effect of MWCNTs on adhesive efficiency was studied using a lap shear test. In order to obtain test specimens, two aluminum plates were pressed together at 135 °C, with neat coPA and coPA + MWCNTs between them. The thickness of the adhesive bond was kept at 0.1mm, and the dimensions of the specimens are shown in Figure 1. The lap-shear test was performed using an MTS810 servo-hydraulic testing machine in accordance with ASTM D 1002. The crosshead speed was kept at 1.3 mm/min.

## 3. Results

### 3.1. Microstructure

The dispersion and distribution of MWCNTs in the coPA matrix were analyzed in terms of agglomerates (so-called macrodispersion) using a light optical microscope. Sample images are presented in Figure 2a,c. At first glance, it can be seen that the coPA containing 5 wt% of MWCNTs (Figure 2a) has more agglomerates than the masterbatch with 10 wt% of MWCNTs (Figure 2c). The higher MWCNT concentration causes a higher shear force during extrusion and more effective destruction of the primary agglomerates. The quantitative analysis confirms this because the percentage ratio of the MWCNT agglomerates in the nanocomposites equals A_A_ = 5.8 ± 1.1% and AA = 3.1 ± 1.0% for 5 wt% and 10 wt% MWCNTs, respectively. The images obtained by TEM are shown in Figure 2b,d present a more detailed analysis of the nanocomposite microstructure. Here, the MWCNTs are visible as single long tubes, loosely connected at some places in the form of bundles. Moreover, they are not arranged in any specific direction. Although MWCNTs are homogeneously dispersed in the copolyamide matrix, there are some empty places where nanotubes do not occur. Such sites are visible as white areas, and they are responsible for decreasing the electrical conductivity of the materials. However, the overall state of MWCNTs dispersion can be classified as good enough.

### 3.2. Rheological Properties

Oscillatory rheology measurements were performed to examine the effect of the addition of MWCNTs on the viscoelastic properties of coPA. As shown in Figure 3 the complex viscosity of pure coPA increases by about 6 orders of magnitude in the presence of 5 and 10 wt% MWCNTs which is related to the restriction of the polymer chains’ movement. It should be noted that there is a negligible difference between 5 and 10 wt% concentration, because a stronger effect is usually visible at low concentrations of CNTs. Here, at 5 wt% the network has already been percolated and further addition of MWCNTs does not change that structure much and does not affect the chains’ movement. Such a high increase in viscosity demonstrates the interaction between coPA and nanotubes. Looking into the curves, it can be seen that since neat coPA is a non-Newtonian liquid, its viscosity is not dependent on the frequency. In contrast, nanocomposites behave as Newtonian liquid with the viscosity decreasing together with the frequency [24]. Interestingly, the viscosity of the materials does not decrease at higher temperatures because the curves remain almost unchaged at 180 °C, 200 °C and 220 °C.

The formation of a percolated network in the coPA-based nanocomposites is also demonstrated by an increase in storage and loss modulus, which describe the viscous and elastic properties, respectively. As can be seen in Figure 4 both moduli increase after the addition of 5 wt% and 10 wt% MWCNTs in comparison to neat coPA. Similarly to the viscosity, there is a negligible difference between 5 and 10 wt% MWCNTs and, what is more, frequency has no effect on both moduli. It should be also noticed that for neat coPA the loss modulus is higher than the storage modulus within the whole frequency range, however for nanocomposites, this is reversed. This shows that nanocomposites behave more like elastic than viscous material, which has been reported for many thermoplastic nanocomposites modified with CNTs [25].

### 3.3. Thermal Properties

The effect of the addition of MWCNTs on the thermal stability of coPA was studied by TGA. The obtained results are shown in Table 2, while the curves are shown in Figure 5. It can be seen that for neat coPA the decomposition which corresponds to 5% weight loss (T_5%_) starts at 310 °C. In the presence of 5 wt% MWCNTs, T_5%_ increases to 336 °C. This behaviour of the material indicates that the presence of CNTs delays the degradation of the polymer. Interestingly, the presence of 10 wt% MWCNTs causes a slight decrease in T_5%_ compared to 5 wt% MWCNTs. The same trend can be observed for 10% weight loss of the material. The maximum degradation temperature for pure coPA occurs at about 440 °C. For material with the addition of 5 wt% MWCNTs, this temperature is 465 °C. It can be concluded that a high MWCNTs content in the coPA structure has less effect on the thermal stability of the polymer than a lower concentration. In comparison, Mahmood.N et al. indicated that adding only 0.5 wt% MWCNTs to the PA6 matrix caused a shift in the temperature distribution by 70 °C [26].

The heating and cooling curves of unfilled coPA and filled with 5 wt% and 10 wt% MWCNT are displayed in Figure 6. From the obtained first heating curves (Figure 6a), two characteristic points, the glass transition temperature and the melting temperature was observed. The addition of MWCNTs does not change T_m_ relative to neat coPA; other researchers have previously observed a similar relationship in polyurethane-based hot melt adhesives [27]. Moreover, the presence of two glass transition peaks in neat coPA indicates that studied adhesive consists of copolymer having segments of PA 6 and PA66 (also confirmed by FTIR analysis). Most likely, the lower T_g_ value is related to the PA6 segment, while the higher value is related to the presence of the PA66 segment [28]. The addition of 5 wt% MWCNTs shifted the second T_g_ peak toward higher values of about 15 °C. This confirms a good dispersion of nanotubes in the coPA matrix, which limits the polymer chains’ mobility [29]. The second heating curve showed similar behaviour of all tested materials (Figure 6b). However, from cooling curves (Figure 6c), the formation of a more marked crystallization peak is observed, shifted towards higher temperatures after the addition of 5 wt% MWCNTs. Introducing MWCNTs in the polymer matrix induces a nucleation process, which affects the crystalline phase formation. These observations were confirmed by other researchers [30]. In the case of coPA with 10 wt% MWCNTs, no further temperature changes were observed. The formation of a new crystalline phase can also be noticed by the changing character of melting curves as well as decreasing the value of enthalpy of melting.

### 3.4. Electrical Properties

Electrical conductivity is the main property since the examined materials based on hot melt coPA and MWCNTs are a candidate for use as ECA’s materials. Figure 7 shows the maps of the conductivity of masterbatches measured on hot-pressed square plates divided into small pieces. It can be seen that the conductivity is not the same at every point of the plate, however, the deviation is not significant (except one value for 10 wt% MWCNTs). The average electrical conductivity value for coPA with 5 wt% MWCNTs was 0.57 ± 0.09 S/m, while introducing 10 wt% MWCNTs increased the conductivity to 3.38 ± 1.07 S/m. In comparison to polyolefine hot melt adhesive containing the same type of MWCNTs, the electrical conductivity at 5 wt% was 0.01 S/m, which is much lower than that obtained for coPA nanocomposites [22]. Obviously, the achieved conductivity is much lower than reported for typical ECAs containing silver as a conductive filler, but the manufacturing process is much easier and faster. The electrical conductivity of neat coPA was below the measuring range of the instrument and was <10^−9^. These results were correlated with the microstructure presented in Section 3.1. The calculated value of A_A_ affects the value of electrical conductivity. Electrical conductivity increases while the average area of MWCNT agglomerates decreases [4,31]. The higher electrical conductivity with lower agglomerate size was the result of better dispersion of the filler in the polymer matrix, and thus more conductive pathways present in the test material.

### 3.5. Mechanical Properties

Figure 8 shows the characteristic stress-strain curves of neat coPA and its composites with MWCNTs. The elastic modulus, tensile strength and elongation at break as a function of the concentration of MWCNTs are presented in Table 3.

Young’s modulus increased with the addition of MWCNTs at a concentration of 5 wt% by 205% and at a concentration of 10 wt% by 330%. Similarly, tensile strength improved by 129% at 5 wt% concentration and by 171% with addition of 10 wt% MWCNTs. Only elongation at break decreased, what was expected and reported already for polyolefine hot melt adhesives containing 5 wt% of MWCNTs [22]. The addition of MWCNTs increases the stiffness and strength of thermoplastic adhesive-based materials caused by the uniform dispersion of nanotubes in coPA matrix. There is also a visible effect of MWCNTs on the Shore D hardness, that grown from 54° ShD for neat coPA to 60 and 64° ShD for 5 wt% and 10 wt% MWCNTs, respectively; such a reinforcing effect has also been reported for single-walled carbon nanotubes mixed with rubber [32].

### 3.6. Contact Angle Measurements

The main property of the ECAs is high adhesion to different substrates. Therefore, the effect of MWCNTs inclusion into coPA hot melt was analyzed by the measurement of the contact angle. The results (Figure 9) show that the average contact angle of neat coPA was 92°, but in the presence of 5 and 10 wt% MWCNTs (lack of differences between the concentration) it was decreased to 80°. This means that MWCNTs change the hydrophobic character of coPA hot melt to a hydrophilic one. This is related to the modification of the surface of coPA due to the incorporation of MWCNTs. Nevertheless, the contact angle of 80° means that the nanocomposites have hydrophilic properties, so their wettability should be improved. Similar results were reported for the other types of coPA hot melt adhesives [23].

### 3.7. Adhesive Properties

From the practical point of view, the main challenge is to keep the same or even better adhesive properties of ECAs after the addition of any conductive fillers. To see how MWCNTs affect the adhesive properties of coPA, a lap shear test was performed. The calculated shear strength is presented in Table 4. For the neat coPA hot melt, the shear strength was 8.4 MPa. Adding 5 wt% and 10 wt% MWCNTs decreased shear strength to 5.4 MPa and 4.8 MPa, respectively. In other words, a high concentration of MWCNTs reduces the adhesive properties of the coPA-based ECAs, due to too much carbon. The literature indicates that up to 3 wt% of MWCNTs, the shear strength increases, but further addition of the filler leads to decreasing the adhesive strength [22]. Similar decreasing in the adhesive properties was found for PA6 hot melt adhesive mixed with lithium bromide, which was explained by lack of the bond formation between the adhesive and the aluminum substrate [33]. It is noteworthy, however, that the shear strength values obtained for coPA hot melt modified by even 10 wt% MWCNTs are high and comparable with the literature [34]. The specimens after lap shear test shown in Figure 10 indicate that in all cases, there was adhesive failure between the adherent and the hot melt adhesive. Moreover, the adhesion strength of the tested materials is influenced by many other properties, including surface roughness of the adherent, wettability of the adherent by adhesive, layer thickness, temperature applied during bonding and even concentration of the hot melt in the bonding layers [35]. These factors will be studied in the future.

## 4. Conclusions

This paper describes a group of electrically conductive adhesives fabricated by melt-blending thermoplastic copolyamide hot melt containing 5 wt% and 10 wt% MWCNTs. The neat coPA was characterized by low viscosity, which, together with the storage modulus and loss modulus, was increased after MWCNTs additions by about 5-6 orders of magnitude due to strong interactions between the coPA chains and nanotubes. At elevated temperatures, the viscosity does not drop because the movement of the polymer chains is hindered due to the high content of MWCNTs. Such behavior is also confirmed by shifting the glass transition temperature of neat coPA by about 15 °C towards higher values at both MWCNT concentrations. However, the nanotubes have a negligible effect of on the melting point, but as the melting curve changes, it was assumed that MWCNTs work as nucleation agents leading to the formation of a new crystal phase. From microscopic images, the area ratio of MWCNT agglomerates calculated by ImageJ was A_A_ = 5.8% for coPA + 5 wt% MWCNTs and A_A_ = 3.1% for coPA + 10 wt% MWCNTs. Fewer agglomerates resulted in higher electrical conductivity for 10 wt%–3.38 S/m, while for 5 wt%, it was 0.58 S/m. The addition of MWCNTs improved thermal stability and temperature of decomposition by about 25–30 °C compared to neat coPA indicates the homogenously dispersed nanotubes. Elastic modulus, tensile strength, and hardness were improved in the presence of MWCNTs by a maximum of 330%, 171%, and 20%, respectively. The addition of MWCNTs also changed the hydrophobic character of coPA hot melt to a hydrophilic one, signifying higher wettability. However, it did not affect the adhesion strength, which decreased from 8.5 MPa to 4.8 MPa as the effect of MWCNT addition. Nevertheless, such adhesion can classify coPA filled with MWCNTs as a good adhesive that, together with achieved electrical conductivity, can be applied as ECAs for EMI shielding applications.

## Figures and Tables

**Figure 1 polymers-14-04371-f001:**
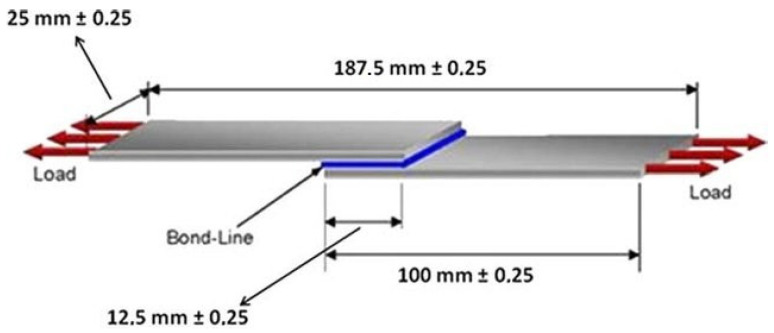
The dimension of the specimen for the lap-shear test [11].

**Figure 2 polymers-14-04371-f002:**
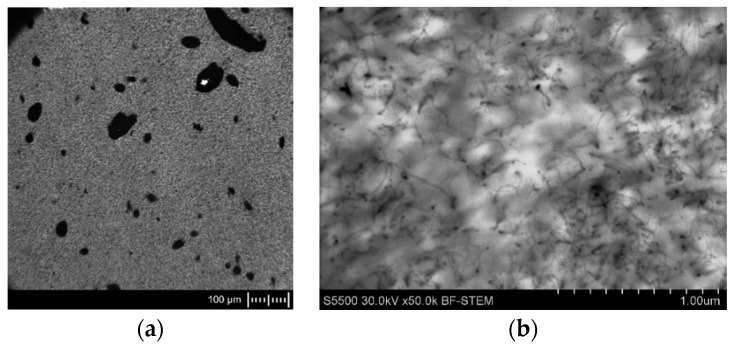
The dispersion of MWCNTs in the coPA: (**a**) macrodispersion in coPA + 5 wt% MWCNTs; (**b**) macrodispersion in coPA + 10 wt% MWCNTs; (**c**) dispersion in coPA + 5 wt% MWCNTs; (**d**) dispersion in coPA + 10 wt% MWCNTs. Scale: 100 µm (**a**,**c**) and 1 µm (**b**,**d**).

**Figure 3 polymers-14-04371-f003:**
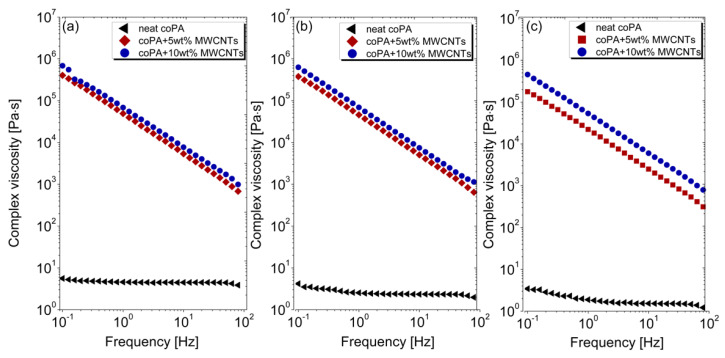
Variation in a complex viscosity as a function of frequency of unfilled and filled coPA measured at: (**a**) 180 °C; (**b**) 200 °C and (**c**) 220 °C.

**Figure 4 polymers-14-04371-f004:**
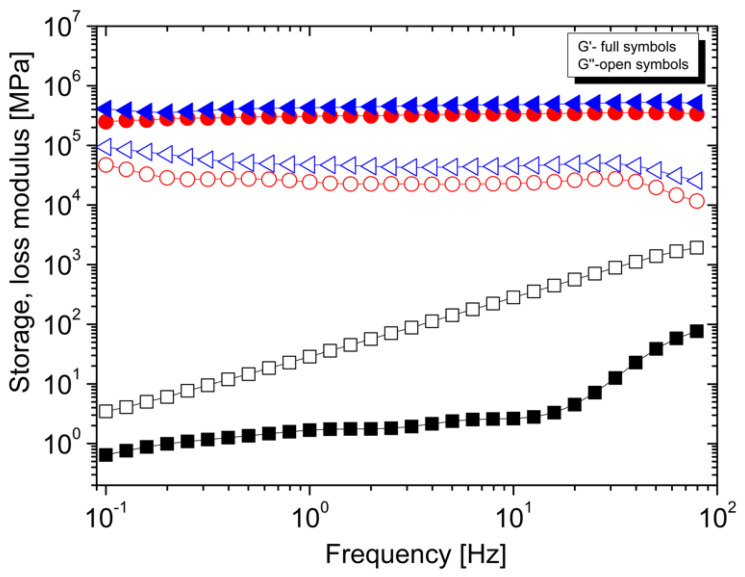
The dependence between storage modulus (full symbols) and loss modulus (open symbols) for neat coPA (black curves), coPA + 5 wt% MWCNTs (red curves) and coPA + 10 wt% MWCNTs (blue curves). Test temperature 180 °C, strain 0.1%.

**Figure 5 polymers-14-04371-f005:**
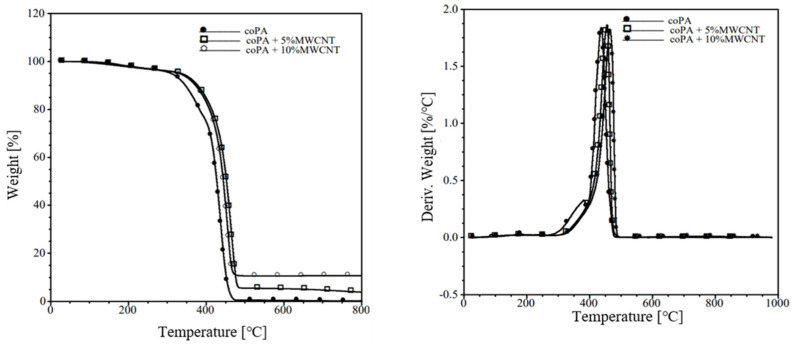
The TGA curves for the neat coPA and its masterbatches containing 5 wt% and 10 wt% MWCNT.

**Figure 6 polymers-14-04371-f006:**
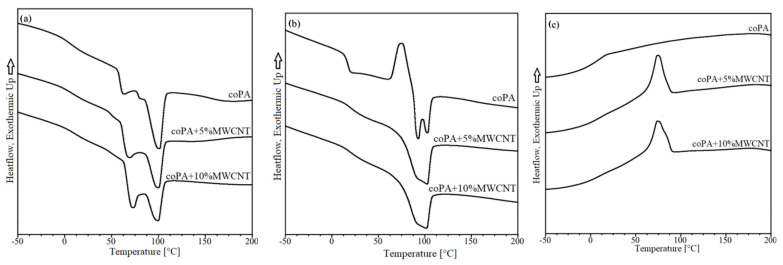
(**a**) First heating curves; (**b**) second heating curves; and (**c**) cooling curves for the unfilled coPA and masterbatches containing 5 wt% and 10 wt% MWCNTs.

**Figure 7 polymers-14-04371-f007:**
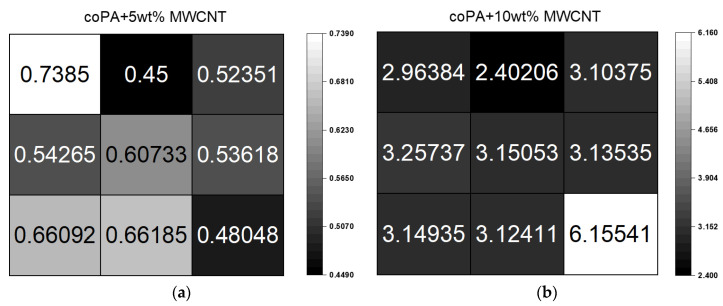
Volume electrical conductivity for masterbatches containing (**a**) 5 wt% and (**b**) 10 wt% MWCNTs.

**Figure 8 polymers-14-04371-f008:**
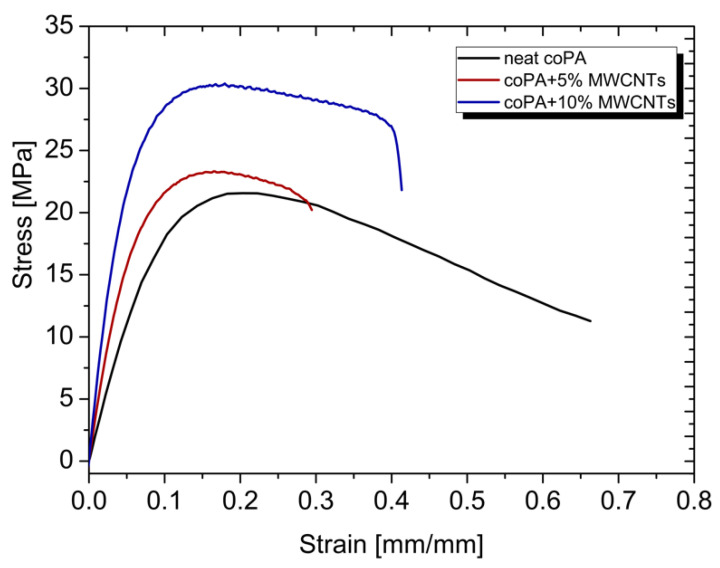
Representative stress-strain curves of coPA and its masterbatches containing 5 wt% and 10 wt% MWCNT.

**Figure 9 polymers-14-04371-f009:**
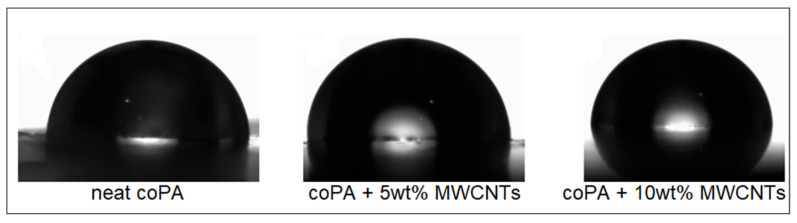
Images taken during contact angle measurement showing the effect of MWCNTs addition on the wettabily of coPA hot melt.

**Figure 10 polymers-14-04371-f010:**
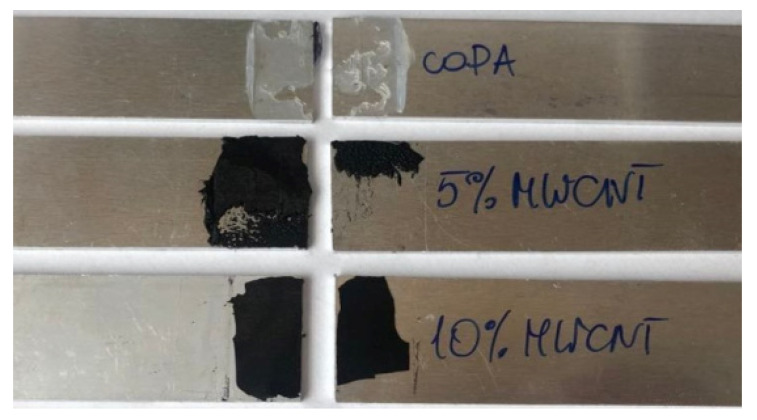
Fracture pattern of joints bonded with MWCNTs modified hot melt at different concentrations.

**Table 1 polymers-14-04371-t001:** Injection moulding parameters for coPA and it’s masterbatches containing 5 wt% and 10 wt% MWCNT.

MWCNTs Content	Barrel Temperature (°C)	Mould Temperature (°C)	Injection Pressure (bar)	Injection Time (s)	Post Pressure (bar)	Post Time (s)
0	110	35	650	3	400	6
5 wt%	150	35	800	5	700	10
10 wt%	150	35	800	5	700	10

**Table 2 polymers-14-04371-t002:** The results of thermal analysis.

TGA						DSC					
Material	T_5%_ (°C)	T_10%_ (°C)	T_d_ (°C)	1st Heating			2nd Heating				Cooling
				T_g_ (°C)	T_m_ (°C)	ΔH_m_ (J/g)	T_g_ (°C)	T_m_ (°C)	T_c_ (°C)	ΔH_m_ (J/g)	T_c_ (°C)
coPA	310	346	437	17 48	101	18	17	103	76	27	-
coPA + 5 wt% MWCNT	336	378	465	18 63	101	13	28	103	-	17	74
coPA + 10 wt% MWCNT	330	373	456	19 66	100	10	30	101	-	15	74

**Table 3 polymers-14-04371-t003:** Mechanical properties of neat coPA and coPA/MWCNTs composites.

Tensile Test				Hardness Test
Material	Elastic Modulus (GPa)	Ultimate Tensile Strength (MPa)	Elongation at Break (%)	Hardness (°ShD)
coPA	0.20 ± 0.02	17.7 ± 1.81	55.9 ± 21.8	54
coPA + 5 wt% MWCNT	0.41 ± 0.02	22.9 ± 0.61	36.7 ± 11.2	60
coPA + 10 wt% MWCNT	0.66 ± 0.03	30.4 ± 0.73	28.6 ± 4.28	65

**Table 4 polymers-14-04371-t004:** The result of the lap-shear test.

Material	Lap Shear Strength (MPa)
coPA	8.4 ± 0.51
coPA + 5 wt% MWCNT	5.4 ± 0.68
coPA + 10 wt% MWCNT	4.8 ± 0.25

## Data Availability

All data are included in the manuscript.

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
