# Peer review of "Electrically Conductive Adhesive Based on Thermoplastic Hot Melt Copolyamide and Multi-Walled Carbon Nanotubes"

_polymers, 2022, doi:10.3390/polym14204371_

Round 1

Reviewer 1 Report

In the article “Electrically conductive adhesive based on thermoplastic hot melt copolyamide and multi-walled carbon nanotubes” by Paulina Latko-DuraÅ‚ek, MichaÅ‚ Misiak, Anna Boczkowska, the effect of Multi-walled carbon nanotubes on the properties of thermoplastic copolyamide was studied. This research work and the results may be interested by broad range of readers. After careful consideration, the manuscript is recommended for publishing in the journal Polymers after addressing following comments:

- In the introduction, give a transcript of CNT, EMI, MWCNT.

- Why did you decide to take such a number of nanotubes? Have you tried adding more?

Author Response

Reviever 1

Q1. In the introduction, give a transcript of CNT, EMI, MWCNT.

A1. Corrected as requested by the Reviewer.

Q2. Why did you decide to take such a number of nanotubes? Have you tried adding more?

A2. The amount of carbon nanotubes we used (7wt%) was selected above the percolation threshold. After that, we tried to use more such as 10wt% but we observed a negative effect on the adhesion properties. Increasing the CNT content is possible, but the work is in progress.  

Reviewer 2 Report

Comments:

1.      How the higher agglomeration and lower agglomeration in your experiments affect the electrical conductivity? Discuss in few lines in section 3.4.

2.      The section 2 is very confusing for the reader. Divide the section 2 into further three subsections namely: 1. Materials 2. Fabrication 3. Characterization technique. To make it better understand for the reader.

3.      The picture quality is very low for all the graphs. Improve the picture quality of all figures (figure 1, figure 3, figure 4, figure 5, figure 6, figure 8) which are demonstrating the experimental results. Use the .emf format (enhanced meta file).

4.      The numbering of table 2 is wrong in the text of section 3.2 at line 266.

5.      The literature work can be updated to most recent published papers. Some suggested references, doi: 10.1109/CEIDP47102.2019.9009938, https://doi.org/10.1002/app.49715, https://doi.org/10.1016/j.compositesb.2020.108204

Author Response

We would like to thank the Reviewer for our article's thorough substantive and editorial evaluation, the positive feedback, and critical comments. Observations will eliminate any shortcomings that may have appeared in the manuscript. They are essential guidelines for improving the quality of future research work. Below we enclose the replies to the comments and recommendations made by the Reviewer.

Q1. How the higher agglomeration and lower agglomeration in your experiments affect the electrical conductivity? Discuss in few lines in section 3.4

A1. The correlation between agglomerate size and electrical conductivity was presented in Section 3.4 at Lines 328-333.

Q2. The section 2 is very confusing for the reader. Divide section 2 into further three subsections namely: 1. Materials 2. Fabrication 3. Characterization technique. To make it better understand for the reader.

A2. Corrected as requested by the Reviewer.

Q3. The picture quality is very low for all the graphs. Improve the picture quality of all figures (figure 1, figure 3, figure 4, figure 5, figure 6, figure 8) which are demonstrating the experimental results. Use the .emf format (enhanced meta file).

A3. The use of the .emf format did not result in improved figures quality. All figures showing the results are from the results processing programs. The highest possible image quality is used.

Q4. The numbering of table 2 is wrong in the text of section 3.2 at line 266.

A4. Corrected as requested by the Reviewer.

Q5. The literature work can be updated to most recent published papers. Some suggested references, doi: 10.1109/CEIDP47102.2019.9009938, https://doi.org/10.1002/app.49715, https://doi.org/10.1016/j.compositesb.2020.108204

A5. Most of the papers cited in our manuscript come from the latest years, most from 2020-2022. The suggested papers are not linked to hot melt adhesives, describing functionalized carbon nanotubes or different fabrication methods. We tried to compare the same manufacturing methods and CNT types since they affect the electrical conductivity and other properties. 

Reviewer 3 Report

Recommendation: Minor revisions needed.

Comments:

The paper by a Latko-DuraÅ‚ek et al. contributes to an analytical approach to assessing the make them electrically conductive, a copolyamide-based hot melt adhesive was mixed with 5 and 10wt% of carbon nanotubes using a melt-blending process. The article gives an interesting scientific perspective ons solvent-based adhesives. I suggest this paper accepted in its present form.

Author Response

Thank you for reading our manuscript and agreeing to publish it in the journal of Polymers.

Round 2

Reviewer 2 Report

The authors have addressed the required comments. I have not more suggestions.